# Thermal Drift Investigation of an SOI-Based MEMS Capacitive Sensor with an Asymmetric Structure

**DOI:** 10.3390/s19163522

**Published:** 2019-08-12

**Authors:** Haiwang Li, Yanxin Zhai, Zhi Tao, Yingxuan Gui, Xiao Tan

**Affiliations:** 1School of Energy and Power Engineering, Beihang University, Beijing 100191, China; 2National Key Laboratory of Science and Technology on Aero Engine Aero-Thermodynamics, Beijing 100191, China; 3The Collaborative Innovation Center for Advanced Aero-Engines of China, Beijing 100191, China

**Keywords:** MEMS, capacitive accelerometer, asymmetric structure, equivalent expansion ratio

## Abstract

High-precision, low-temperature-sensitive microelectromechanical system (MEMS) capacitive accelerometers are widely used in aerospace, automotive, and navigation systems. An analytical study of the temperature drift of bias (TDB) and temperature drift of scale factor (TDSF) for an asymmetric comb capacitive accelerometer is presented in this paper. A five-layer model is established for the equivalent expansion ratio in the TDB and TDSF formulas, and the results calculated by the weighted average of thickness and elasticity modulus method are closest to the results of the numerical simulation. The analytical formulas of TDB and TDSF for an asymmetric structure are obtained. For an asymmetric structure, TDB is only related to thermal deformation and fabrication error. Additionally, half of the fixed electrode distance is not included in the expressions of Δd and ΔD for asymmetric structures, thus resulting in the TDSF of the asymmetric structure being smaller compared to a symmetric structure with the same structural parameters. The TDSF of the symmetric structure is [−200.2 ppm/°C, −261.6 ppm/°C], while the results of the asymmetric structure are [−11.004 ppm/°C, −72.404 ppm/°C] under the same set of parameters. The parameters of the optimal asymmetric structure are obtained for fabrication guidance using numerical methods. In the experiment, the TDSF and TDB of the packaged structure and the non-packaged structure are compared, and a significant effect of the package on the signal output is found. The TDB is reduced from 3000 to 60 μg/°C, while the TDSF is reduced from 3000 to 140 ppm/°C.

## 1. Introduction

Microelectromechanical systems (MEMS) have become highly promising systems in the 21st century. Due to their miniaturization, variety, and stability, MEMS fabrication is widely used in the manufacture of accelerometers, gyroscopes, pressure mano-meters, etc. [1]. In MEMS sensor systems, microaccelerometers play important roles in products such as automotive systems and navigation systems and are gradually being used in aerospace and other high-tech fields [2]. Particularly, the capacitive accelerometer has the unique advantages of simple processing, high yield, a low temperature coefficient, low noise, and high sensitivity [3,4]. Therefore, the development of capacitive accelerometers has become a key research object, both at home and abroad. Through the improvement of processing technology [5], circuit matching [6], and optimization of structural design [7,8], the influence of thermal noise [9,10,11,12] of the capacitive accelerometer can be reduced, and the sensitivity of the device can be improved.

However, the accuracy of the acceleration sensor is greatly affected by the temperature. When the temperature conditions are poor, a high temperature coefficient [13], low linearity [14,15], and temperature drift [16] are extremely common. Therefore, research on high-precision, low-temperature sensitivity MEMS accelerometers has received great attention [17,18,19,20,21,22,23,24]. The temperature drift of bias (TDB) and temperature drift of scale factor (TDSF) are the two main indicators that characterize the effects of thermal phenomena on accelerometers. The analytical formulas for the TDB and TDSF of the symmetric comb capacitive accelerometer were obtained through a theoretical analysis in a previous study [18], and the temperature compensation structure was designed to reduce the TDB and TDSF of the accelerometer. A previous study [19] obtained a low temperature coefficient and high linearity by improving the sensor structure and explored the theoretical relationship between the structure and linearity of the device. After determining the specific structure, dimension, and packing method, the values of TDB and TDSF were obtained [18,19], so these two parameters can be used to guide the theoretical design of the device. He. J et al. obtained a TDB of 179 μg/°C and a TDSF of −9.8 ppm/°C through the design of the structure [19]. S. Schröder et al. used a two-sided wire bond method to eliminate the original stress and improved the reliability of the device [20]. Zhang. Y et al. adopted a four-point support method to reduce the contact area and obtained a linearity of 0.435% and a TDB of 0.02%/°C [21]. Geisberger. A et al. used a centralized positioning method to obtain good linearity and a TDSF in the range of −40 to 125 °C with a drift of 0.3% [22]. H. Ko et al. adjusted the output signal and actual temperature to the ideal temperature output. In the range of −40 to 125 °C, the zero-drift error changed from 16.1 to 135.2 μg, and the equivalent noise changed from 93.5 to 514.0 μg/$\sqrt{\mathrm{Hz}}$ [23]. C. Falconi et al. accurately controlled the hardware temperature in the range of 0~120 °C, and the temperature control accuracy reached ±0.1 °C [24].

In this work, a structural model of the asymmetric comb capacitive accelerometer is proposed. The calculation of the equivalent expansion ratio in the existing analytical formula is analyzed and brought into the model of the asymmetric structure. Then, the theoretical analysis formula of the asymmetric structure is obtained. The differences between the TDB and TDSF of symmetric structures and asymmetric structures are discussed. Additionally, the optimal design dimension of the asymmetric structure is obtained by a numerical method, and the structure is checked to prevent adhesion. Finally, the experiments are carried out. The designed structure is processed, and the thermal effect of the accelerometer is compared between the packaged condition and the non-packaged condition.

## 2. Thermal Effect Analysis of an Asymmetric Structure

### 2.1. An Asymmetric Structure Model and Detection Principle

The structure of the symmetric comb capacitive accelerometer was modeled, and the model was used to analyze the thermal deformation. The analytical formulas related to temperature drift were obtained for TDB and TDSF. The symmetrical structure was characterized in that a moving electrode and a fixed electrode which are bilaterally symmetric with each other are distributed at both ends of the proof mass. The structure studied in this paper is similar to the symmetrical structure, with a capacitive structure composed of asymmetric moving electrodes and fixed electrodes formed at both ends of the mass. This kind of asymmetric structure has good intrinsic series accumulation in the theoretical analysis process, which will be described in detail later. The asymmetric structure shown in Figure 1 is composed of sensing mass elements, fixed electrodes, and substrates. Sensing mass elements include the proof mass and multiple sets of moving comb and elastic beam structures. The moving combs of this accelerometer structure are extended to both sides of the proof mass to form a double-sided comb structure, and the elastic beams at both ends of the proof mass are connected to the anchors of the proof mass. The proof mass and the moving electrodes are arranged upon the substrates. Every moving comb of the proof mass is a moving electrode of the capacitor, forming a differential capacitor with every fixed comb. The fixed combs are single-side structures that are directly fixed on the substrate. Every moving electrode of the proof mass has different distances from its two adjacent fixed electrodes. The proof mass can move axially along the sensitive direction shown in Figure 1a, obtaining a displacement change. This structure mainly uses the capacitance formed by the side with a narrow sensitive gap to detect the change in the acceleration, ignoring the capacitance of the wide gap.

TDB and TDSF characterize the thermal sensitivity of accelerometers. In principle, the smaller the absolute values of these two indicators are, the better. However, due to the difference between the equivalent coefficient of thermal expansion (CTE) $\alpha_{eq}$ and the CTE of silicon $\alpha_{s}$, it is difficult to reduce the TDSF and TDB to 0. The silicon combs of different structures and the expansion direction of the spring will affect Δd and ΔD in the formula. Therefore, based on the analysis of the symmetrical structure, the change in displacement of the electrode spacing of the asymmetric structure was analyzed to obtain the idea of improving the asymmetric structure.

### 2.2. Analysis of Equivalent CTE

In the process of deriving the theoretical formulas of TDSF and TDB, the concept of $\alpha_{eq}$ is proposed, but there is no specific calculation model and method for $\alpha_{eq}$. Therefore, in this work, before the formula was derived, a new model was established for the calculation of $\alpha_{eq}$. The different CTE values between the substrate and the package causes a CTE mismatch. Considering that the CTE values of the materials of the various parts are different, in order to simplify the calculation, a value considering the CTE of each part can be obtained, which is called the equivalent expansion ratio $\alpha_{eq}$. The fixed electrodes are bonded to the package, and the CTE that equals the equivalent CTE $\alpha_{eq}$ of both the silicon and the package is obtained. However, the moving electrodes and the proof mass are connected to a silicon spring with a small stiffness coefficient, and the spring’s deformation is much larger than that of the moving electrode and the proof mass caused by thermal expansion. Therefore, the CTE mismatch between the package and the moving part can be ignored, and the CTE of the moving electrodes and the proof mass is the CTE of the silicon $\alpha_{s}$ itself. The difference between $\alpha_{eq}$ and $\alpha_{s}$ causes the gaps of the capacitor combs to change, which affects the output signal. In this work, theoretical assumptions and simulations were used to calculate $\alpha_{eq}$. The thickness of SiO_2_ is 1 μm while the thickness of epoxy is 4 μm, so the following formulas ignore the effects of SiO_2_ and epoxy’s thickness, and assume a total thickness of silicon of $h_{2}$ and a ceramic thickness of $h_{1}$. There are four methods to calculate $\alpha_{eq}$, namely the average method, the weighted average of thickness method, the weighted average of elasticity modulus method, and the weighted average method of thickness and elasticity modulus, as follows:(1)$$\alpha_{eq1}=\frac{\alpha_{1}+\alpha_{2}}{2}$$
(2)$$\alpha_{eq2}=\frac{h_{1}\alpha_{1}+h_{2}\alpha_{2}}{2(h_{1}+h_{2})}$$
(3)$$\alpha_{eq3}=\frac{\alpha_{1}+\alpha_{2}}{2}+\frac{\left(E_{1}\alpha_{1}-E_{2}\alpha_{2})(E_{2}-E_{1}\right)}{2E_{1}E_{2}}$$
(4)$$\alpha_{eq4}=\frac{E_{1}h_{1}\alpha_{1}+E_{2}h_{2}\alpha_{2}}{2E_{eq}(h_{1}+h_{2})},E_{eq}=\frac{h_{1}E_{1}+h_{2}E_{2}}{h_{1}+h_{2}}.$$

It is assumed that $E_{1}$, $h_{1}$, and $\alpha_{1}$ represent the elasticity modulus, thickness, and CTE of the ceramic, and $E_{2}$, $h_{2}$ and $\alpha_{2}$ represent the elasticity modulus, thickness, and CTE of silicon. Equation (1) is the average method, and only the arithmetic mean of the two CTEs is considered, while the other physical parameters, such as the thickness and elasticity modulus, are not. Equation (2) indicates the effect of the thickness on $\alpha_{eq}$, ignoring the effect of the elasticity modulus. Equation (3) indicates the effect of the elasticity modulus on $\alpha_{eq}$, ignoring the thickness. In Equation (4), it is assumed that the two materials are free to expand first and then contract or pull out according to the CTE of the two, as shown in Figure 2. The overall expansion relationship between the two materials shown in Figure 2 is complex. However, to simplify the understanding and calculation, this complex expansion relationship was simply divided into two steps. In the first step, the upper and lower layers are first freely expanded, as represented by $\Delta X_{1}$ and $\Delta X_{2}$, respectively. In the second step, assuming the upper layer compresses and the lower layer expands, the upper layer is contracted ($\Delta X_{3}$) and the lower layer is pulled out ($\Delta X_{4}$) to achieve the same final deformation result. The assumption of these two steps can simplify the expansion relationship between the two materials, which is conducive to the establishment of the calculation model.

Equation (4) indicates the effect of the elasticity modulus and thickness on $\alpha_{eq}$. The equivalent elasticity modulus $E_{eq}$ is the weighted average of the thickness, and the overall displacement and the weighted average of the elasticity modulus are added into $\alpha_{eq}$. Since the materials used in the package contain epoxy and SiO_2_, although they are not taken into account in the theoretical calculations, they must be taken into account in the numerical simulation. The parameters are shown in Table 1.

The model used for the numerical simulation is shown as Figure 3. The model is divided into five layers which are composed of silicon, SiO_2_, silicon, epoxy, and ceramic.

In the numerical simulation, finite element analysis was used. A hexahedral mesh with a mesh number of about 400,000 was used. In the numerical simulation process, the thin layer of SiO_2_ in the middle of the SOI (Silicon-on-insulator) was 1 μm thick and the thickness of the epoxy on the bottom surface of the SOI was 4 μm, so the mesh of the two layers was encrypted. However, it is necessary to pay attention to the fact that the thickness of the two layers is not considered in the theoretical analysis, because the influence on the theoretical calculation results is negligible, so the thicknesses of these two layers were ignored in the theoretical derivation. The ambient temperature around the device was set to 70 °C, and the temperature difference from normal temperature was 50 °C. Two assumptions were used in the numerical simulation: (1) Free expansion with no boundary constraints. This is based on the assumption that the ceramic package and the chip were in a high-temperature environment, without circuit board constraints. (2) A fixed support was set on the bottom surface of the ceramic package. This was based on the assumption that the ceramic package and the chip were in a high-temperature environment in the actual working process with circuit board constraints. The simulation results for the two assumptions at a total silicon layer thickness of 335 μm are shown in Figure 4.

Figure 5 shows the stress distribution of the model during free expansion. A large concentration of stress occurred on the edges of the bottom and on the SiO_2_ layer. Figure 6 shows the results of four theoretical calculation methods and two numerical simulation methods. It was assumed that only the thickness of the silicon on the bottom of the SOI changed, and the thickness of the silicon of the top layer and the thickness of the SiO_2_ did not change. The $\alpha_{eq}$ calculated by the average method (Equation (1)) and the weighted average of elasticity modulus method (Equation (3)) did not change with the total thickness of silicon, and the measured values were 4.5 and 7.5 ppm/°C, respectively. The $\alpha_{eq}$ calculated by the weighted average of thickness method (Equation (2)) and the weighted average of thickness and elasticity modulus method (Equation (4)) decreased with an increasing total thickness of silicon, from 5.5 and 3 ppm/°C to 4.5 and 2.7 ppm/°C, respectively. The calculation results of Equations (2) and (9) were similar to the results of the two simulations. Additionally, the result of Equation (4) was between the simulation results of the free expansion model and the fixed support model, although it was closer to the result of the fixed support model. This shows that the calculation result of Equation (4) and the simulation result of the fixed support model are closer to the real result, so the calculation of $\alpha_{eq}$ in this work was based on these two methods. The actual processed silicon wafer thickness was 335 μm. $\alpha_{eq}$ was taken as 3 ppm/°C to simplify the calculation.

The correctness of the theoretical model can be verified by others’ experimental data. The $\alpha_{eq}$ of the glass-silicon model obtained was 4.2 ppm/°C. Using the equivalent expansion ratio model in this paper, the result was 4.16 ppm/°C with a relative error of 1%. The calculation model of $\alpha_{eq}$ in this paper was a model in which SOI is bonded to a ceramic substrate by epoxy. It was assumed that atomic-level bonding or some inseparable contact formed between the silicon and SiO_2_. The package model of the symmetric structure assumes the direct bonding of the glass to the ceramic substrate after the anodic bonding of silicon and glass, so the packaging modes of the sensitive components of the two are essentially the same. This consistency can also be used to verify the results of the model calculations in this paper and the existing literature.

### 2.3. Analytical Formulas for TDB and TDSF of Asymmetric Structure

In Equation (5), there are no factors affected by structural changes during the expression derivation process of TDB, so the expression of TDB did not change in this work. Physically, TDB refers to the temperature drift caused by temperature change, which is mainly caused by machining errors and thermal deformation. Theoretically, although in the asymmetric structure, the influences of the machining error and thermal deformation do not change, Equation (5) can also be applied to the asymmetric structure:(5)$$TDB=\frac{p_{0}|{}_{\Delta T}}{\Delta T}\approx \frac{-Ku_{c}}{m\Delta T}=\frac{K_{A}-K_{B}}{m}(\alpha_{eq}-\alpha_{s})L_{m}$$
(6)$$TDSF=\frac{K_{0}|{}_{\Delta T}-K_{0}}{\Delta TK_{0}}=-TCS-\frac{\frac{\beta \Delta d}{d\Delta T}-\frac{\Delta D}{D\Delta T}}{\beta -1}.$$

TDSF characterizes the change in slope of the linear segment of the test, representing the nonlinearity of the test. It can be seen from Figure 1 that the moving comb and the fixed comb of the asymmetric structure expand outwards, but the effect of the expansion of the moving comb on the upper part is to increase the narrow gap d and reduce the wide gap D. On the lower part, the effect of the expansion of the moving comb is to increase the wide gap D and reduce the narrow gap d. This kind of effect is the opposite in the case of the fixed comb. So, the following formulas can be obtained:(7)$$\Delta d_{Ai}=\alpha_{s}iL_{1}\Delta T-\alpha_{eq}(il_{1}-d-2s)\Delta T+u_{c}$$
(8)$$\Delta d_{Bi}=\alpha_{s}(i+1)L_{1}\Delta T+\alpha_{eq}(il_{1}+d+2s)\Delta T-u_{c}$$
(9)$$\Delta D_{Ai}=-\alpha_{s}iL_{1}\Delta T+\alpha_{eq}(il_{1}-d-2s)\Delta T-u_{c}$$
(10)$$\Delta D_{Bi}=\alpha_{s}(i+1)L_{1}\Delta T-\alpha_{eq}(il_{1}+d+2s)\Delta T+u_{c}$$
(11)$$\Delta D_{0}=\alpha_{s}L_{1}\Delta T+\alpha_{eq}(l_{1}-d-2s)\Delta T-u_{c}$$
where $u_{c}$ refers to the displacement of the proof mass, 2s is the width of the comb, $l_{1}$ is the distance between two fixed electrodes, $L_{1}$ is the distance between two moving electrodes (Figure 1), $\Delta D_{0}$ is the change in the wide gap between the two sides of the central axis, and the expansion of the moving comb and the fixed comb makes $\Delta D_{0}$ increase. The average value of the change in gaps can be expressed as
(12)$$\Delta d=\sum_{i}^{N}\frac{d_{Ai}+d_{Bi}}{2N}=(d+2s)\alpha_{s}\Delta T$$
(13)$$\Delta D=\sum_{i}^{N}\frac{D_{Ai}+D_{Bi}+D_{0}}{2N-1}=(D+2s)\frac{\alpha_{eq}+\alpha_{s}}{2}\Delta T.$$

When the series is accumulated, due to the inherent characteristics of the asymmetric structure, partial results may cancel each other out, which will be beneficial to the reduction of TDSF. Ignoring the second order, the change in capacitance value can be expressed as
(14)$$c_{A}\approx \varepsilon_{A}\Omega (\frac{2N}{d+\Delta d+u_{c}+x}+\frac{2N-1}{D+\Delta D-u_{c}-x})$$
(15)$$c_{B}\approx \varepsilon_{A}\Omega (\frac{2N}{d+\Delta d-u_{c}-x}+\frac{2N-1}{D+\Delta D+u_{c}+x}).$$

From Equations (12)–(15) and Equation (6), the analytical formula of TDSF of the asymmetric structure can be expressed as
(16)$$TDSF=-TCS+\frac{1}{\beta -1}[\frac{\alpha_{s}+\alpha_{eq}}{2}-\alpha_{eq}\beta -\beta \frac{2s\alpha_{eq}}{d}+\frac{s(\alpha_{s}+\alpha_{eq})}{d\beta }].$$

Equations (12) and (13) show that Δd and ΔD are proportional to the temperature difference ΔT and their original values d and D, and they are positively correlated with the comb width 2s. Compared with the formulas of the symmetric structure, half of the fixed electrode distance is not included in the expressions of Δd and ΔD of the asymmetric structure. This means that the asymmetric structure eliminates the effect of the electrode distance, which will make the TDSF value of the asymmetric structure one order of magnitude smaller than the symmetrical structure.

Using the material properties and structural dimensions stated in [18], under the same conditions, the TDSF calculation result of the symmetric structure was [−200.2 ppm/°C, −261.6 ppm/°C], and the calculation result of the asymmetric structure was [−11.004 ppm/°C, −72.404 ppm/°C]. This shows that under the same structural dimensions and material properties, the asymmetric comb structure design is more reasonable than the symmetrical structure.

### 2.4. Optimal Dimensions for Asymmetric Structure

In accordance with the conclusions presented in Section 2.3, by using Equation (16) as the design principle combined with the processing conditions of the laboratory and common design experience parameters for the optimization of structural design, suitable processing conditions were obtained. Meanwhile, the optimal structural size parameters of the actual situation were obtained.

The ratio of $\beta \frac{2s\alpha_{eq}}{d}$ and $\frac{s(\alpha_{s}+\alpha_{eq})}{d\beta }$ is much larger than 1, so the theoretical analysis was able to ignore $\frac{s(\alpha_{s}+\alpha_{eq})}{d\beta }$. Assuming a TDSF of 0, the following formula was obtained:(17)$$\frac{s}{d}=\frac{1}{2\alpha_{eq}}[\frac{\left(\alpha_{s}+\alpha_{eq}\right)}{d\beta }-\alpha_{s}-\frac{\beta -1}{\beta }TCS].$$

Figure 7 was drawn according to Equation (17), where s/d is the dependent variable and β is the independent variable.

It can be seen from the Figure 7 that as s/d increases, β increases. Starting from β=1, the curve grows very fast, and it gradually becomes balanced later. Equation (16) shows that TDSF is a function of β, s and d. Therefore, the numerical method “vertical and horizontal halver method” (Figure 8) can be used to find the zero point of Equation (19).

Since the electrode spacing of the comb structure is too small, the electrodes will adhere together, resulting in structural failure. In order to prevent adhesion, the value of d was selected to be 10.

In summary, in order to find the smallest TDSF, the zero-point was selected by the vertical and horizontal halver method. Based on the processing capability and the selection of the empirical parameters, the size of the optimized version of the asymmetric structure was determined to be β=5,s=15,d=10. The optimized version of the design drawing is shown in Figure 9. The white part is to be etched, and the black part is to be reserved.

The structural dimensions of the optimized design are shown in Table 2. The previous design process mentioned the problem of structural failure caused by the adhesion effect. Therefore, after the structural design was finished, it was necessary to check whether the electrodes would adhere together. The purpose of this was to verify whether the narrow gap *d* = 10 met the minimum requirements for preventing adhesion. Raccurt O. et al. [25] unified the calculation of two different spacings and carried out experimental verification and found that a reasonable data curve would be obtained according to the weighting method. As long as the following two dimensionless parameters are greater than 1, respectively, the requirement can be achieved.
(18)$$N_{EC}=\frac{N_{EC1}\times N_{EC2}}{N_{EC1}+N_{EC2}}$$
(19)$$N_{P}=\frac{N_{P1}\times N_{P2}}{N_{P1}+N_{P2}}$$
where $N_{EC1}$ and $N_{EC2}$, respectively, represent the liquid-adhesive dimensionless parameters of two different directional spacings, $N_{EC}$ is the overall influence after weighting, $N_{P1}$ and $N_{P2}$ respectively represent the dimensionless adhesion parameters caused by the electrostatic forces of two different directional spacings, and $N_{P}$ is the overall influence after weighting. In this work, assuming β = 5, according to the above formulas, the trend graph of the dimensionless parameters with narrow gap d was obtained, as shown in Figure 10.

The curves of the $N_{P}$ of the four different solutions are coincident, which is due to the fact that the solutions were applied to the silicon base, and the curves increase with the increase of *d*. In order to ensure that the electrodes do not adhere together, *d* must be greater than 8, in order to ensure $N_{P}$ > 1. The curves of the $N_{EC}$ of the four different solutions are not coincident, and they increase with the increase of *d*. In order to ensure that the electrodes do not adhere together, *d* must also be greater than 8, in order to ensure $N_{EC}$ > 1.

## 3. Experiment

### 3.1. Fabrication Process

According to the optimized size of the asymmetric structure obtained in Section 2.4, the accelerometer was processed. The structure of the MEMS capacitive accelerometer is precise. The minimum size must be 5 μm or less, the maximum aspect ratio must be 5 or more, and the verticality must be 90°. At the same time, the thickness of each beam is below 30 μm. The most difficult process is to deeply etch the bulk silicon structure using inductively coupled plasma (ICP) to get a high aspect ratio. At present, high-precision deep etching methods have been applied, and high aspect ratio structures [26] and shaped three-dimensional structures [27,28] can be obtained by etching. The literature [26] completed the optimization of the parameters of the ICP etching process through a large number of contrasting experiments and formed the etching experience formula of single crystal silicon structures of different widths and depths. In this work, four fabrication schemes were designed for the SOI-based accelerometer to finally obtain the finished product. Table 3 and Figure 11 below show the process flow.

As can be seen from the specific process flow, the first step was to etch the structure of the back cavity. The main difficulties in this step were controlling the precision of deep silicon etching and removing the oxide layer by rapid corrosion using HF (Hydrofluoric acid). In the second step, the metal electrode was processed. A layer of metallic aluminum was sputtered on the surface by magnetron sputtering. It should be noted that after corrosion during the second step, the conventional Piranha cannot be used to remove the photoresist and acetone should be used. In the third step, the processing of the comb electrodes was performed. In this step, in order to ensure the safety of the SOI, the entire wafer was protected at the bottom by the handle wafer, and the precise comb electrodes structure was etched by ICP.

In the process of fabrication, there is a problem of adhesion. As shown in Figure 12, the moving proof mass structure and the bottom structure adhere to each other during processing. In order to avoid this problem, a combination of dry etching using HF and wet etching was adopted, and the suspended structure in the device was successfully released, as shown in Figure 13. In addition, the method of adding a substrate was used to protect the fragile structure, and the etching process avoided the damage caused by ultrasonic vibration in the Lift Off process.

Figure 14 shows the final device processing results. Figure 14a shows the result of the SEM observation. Figure 14b shows the small pattern taken by the camera, and the figure contains the chip with the plated electrode and the chip without the electrode. Figure 14c shows the wafer that has not been etched, and Figure 14d is the frame that has been etched away. After the processing was completed, the electrical signal in the accelerometer was transmitted through the metal pad on the chip and connected to the test circuit.

### 3.2. Testing Process

The test method involved the connection of the chip to the surface of the printed circuit board (PCB) and use of the heating plate to directly heat the bottom surface of the PCB. The purpose was to explore the influences of different heating temperatures on the signal output of the accelerometer under static conditions. For this purpose, the PCB with the chip was placed at different angles to test the voltage outputs of 0 g, ±1 g, and ±0.5 g at different temperatures (20–70 °C). Then, lines of the scale factor drift and the bias drift were obtained through the output result, and finally, TDB and TDSF were obtained by the slopes of these two lines.

The initial method was to attach the chip directly on the PCB without packaging the chip, attach the PCB to the heating plate, and connect the signal to the test circuit through the long wire. However, if the wire was too long, the DuPont line through which the signal passes would be easily affected by the electromagnetic field and the thermal field, which would introduce a large error into the circuit signal output, which could not be removed by simple filtering.

In order to avoid this problem, the chip was packaged to reduce the influences of the external electromagnetic field and the thermal field on the chip signal transmission. The test results were further analyzed after packaging.

### 3.3. Package Method and Circuit Design

There are many ways to package a chip. This work deals with temperature issues, so ceramic was chosen as the substrate shell. The steps of the ceramic shell process were as follows:(1)Tape casting and cutting;(2)Framing via punching, hole filling, and screen printing;(3)Lamination, snapping, block shaping, and co-firing;(4)Ni-plating and Au-plating.

The structure of the package is shown in Figure 15. The electrical signal of the chip was led to the step of the middle layer of the shell through the gold wire and then conducted to the back of the chip through the hole. The chip and the shell were bonded together by organic glue and then cured by a temperature rise of 3 °C/min and a temperature of 150 °C for 4 h.

After the package was completed, the chip was as shown in Figure 16. There are four molded packaged chips in the figure. The back of the chip is in the yellow frame in the upper left corner of the picture, and the picture on the right side of the yellow and red frames is the enlarged chip image. As can be seen in Figure 16, the wires were in good condition, and there were no obvious defects or breaks, indicating that the chip package was successful. Subsequently, the chip needed to be soldered together with PCB according to the mark position. Then, the entire accelerometer sensor system was completed.

A photo of the chip and circuit after packaging is shown in Figure 17. The signal obtained by the oscilloscope was more stable than the previous signal, while other field effects caused less interference on the device. After the package, the ceramic shell isolated part of the electromagnetic interference. Additionally, the wires of the chip were obviously shortened and fixed, so that the electrical signal was relatively pure, and the noise is small. In order to further condition the output signal, the AC bridge method was applied. In this method, the differential capacitor to be tested was placed in a circuit of alternating current, and the change in the charge of the mass in the capacitive acceleration sensor was detected, and then the change in the current was converted into the change in the voltage value, and the voltage signal was the output. The MS3110 is a chip that provides AC variable clock oscillations, and it can convert the sensor’s charge change into a voltage output. In this paper, a readout circuit with an MS3110 chip as the core was designed. The circuit was connected as shown below (Figure 18). The circuit was divided into 8 modules: a 5 V IN power module, a MS3110 chip clock oscillation module, a Signal Out module, a From Sensor module, a 5 V to 16 V voltage conversion module, a To MCU (Microcontroller Unit) module, a +16 V Jumper module, and a screw retention module. The signal output module not only provided the ground signal but also output the analog signal. It also provided a 2.25 V reference voltage for the second pin of the MS3110, as shown in Table 4. This was used as a reference signal for the subsequent analog-to-digital conversion chip. This reference voltage was also used to check whether the MS3110 was working normally.

### 3.4. Experimental Device and Result

The experimental device used in this work is shown in Figure 19 below. The packaged sensor was placed on a rotating table that could be rotated and leveled, and the heating plate was pasted on the bottom of the PCB to heat the sensor. The thermometer for the heating plate was used to measure the operating temperature of the heating plate, and the actual temperature of the sensor’s surface was directly tested using a thermocouple thermometer. The level was placed on the rotary table, the rotary table was adjusted to a horizontal state, the sensor circuit was connected to a voltage of 5 V, the oscilloscope was connected through the wire for data acquisition, and the computer was used to record the data. The heating plate temperature was increased step-by-step from room temperature, with a gradient of 10 °C, so that the actual temperature of the sensor reached 70 °C. The actual temperature of the sensor at each temperature was measured separately, and the voltage signal at this time was collected. Then, the temperature was lowered from high to low to room temperature, and the voltage signals at this time were collected separately. After that, the rotating table was rotated to 30°, 45°, 60°, 90°, 120°, 150°, and 180°, respectively, and the above measurement work was repeated to obtain the changes in the voltage signal with temperature under different acceleration conditions.

After adjusting the test system, it was necessary to test whether the circuit could work normally at a normal temperature and to confirm whether the MS3110 chip was providing the clock signal normally. There were three pins on the output port of the circuit, named GND, Vout, and Vref, which represented the ground, signal output, and reference voltage signals.

If the circuit was working normally, when the oscilloscope channel CH1 was connected to the GND and Vref pins, the oscilloscope would get a stable voltage of 2.25 V. If the circuit was not working properly, it would get a voltage of 0. Regardless of the state of the sensor, the standard voltage should be kept at 2.25 V. After setting the initial state of the sensor, the PCB was erected along the four sides and the reference voltage values were tested. The results in Table 4 show that the reference voltage was not affected by the direction and was in normal working condition. Under the condition of a normally working sensor, the oscilloscope was directly connected to the output end of the sensor, and the static output signal was measured under the state of horizontal placement. The output result of the signal is shown in Figure 20a, and the output signal was very stable. The output value was maintained near 2.23 V. After that, the rotary table power supply was turned on, and the output signal was measured at a low speed. Since the rotation speed was low (the period is 16 s), it can be regarded as a quasi-static process, and the output signal should be a complete sinusoidal AC signal, as shown in Figure 20b. The above test showed that the current accelerometer and the conditioning circuit perform well to output the desired signal and have good stability and linearity. Finally, the experiment was carried out, and the corresponding experimental results were obtained.

As shown in Figure 21, the relationship between the acceleration value and the voltage of the output showed a good linear relationship. The output resolution after packaging was about 40 mv/g, indicating that the acceleration sensor reached a normal state at a normal temperature. As can be seen from Figure 21, the voltage value on the ordinate was the absolute value of the output signal, and the stable voltage value at 0 g was not subtracted. The voltage output after packaging was more regular than before packaging because the curves were almost parallel, and the linearity was better. At high temperatures, the linearity of the packaged accelerometer remained stable. Before packaging, the accelerometer had poor linearity at high temperatures and was more volatile than in the low temperature state. This also shows that after packaging, the overall effect was better than before packaging.

Considering that the influence of temperature cannot be eliminated, this work optimized the size of the accelerometer and obtained the optimized accelerometer output and temperature characteristic curve, as shown in Figure 22. The values of the scale factor drift and the bias drift after packaging were reduced compared to the values before packaging. This is due to the fact that the air wall in the chip has a certain heat insulation effect, the ceramic heat conduction is fast, and the signal wire is shortened.

The TDSF is actually the slope of the line in (a) and (c) above, and the TDB value is the slope of the line in (b) and (d) above. A histogram of each coordinate point in the figure was drawn according to the temperature distribution, as shown as Figure 23. The results of TDSF and TDB are shown in Table 5.

The absolute values of TDB and TDSF decreased by two orders of magnitude after packaging compared with the values before packaging, and the absolute error also decreased. However, the relative stability of the TDB after packaging was not as good as that before packaging, due to the limited accuracy of the device.

## 4. Conclusions

This work proposed the design of a comb capacitive accelerometer with an asymmetric structure. The temperature effect was modeled under static conditions and compared with that of the symmetrical structure. Firstly, the estimation method and mathematical model of the equivalent expansion ratio of the multi-layer structure of the accelerometer were established and verified by numerical simulation. The model with the closest simulation result was adopted to determine the equivalent expansion ratio in this work (3 ppm/°C). Then, according to the theoretical formula derivation process of the TDSF and TDB of the symmetric structure, theoretical derivation of the TDB and TDSF of the asymmetric structure was carried out, and it was found that TDB did not change due to structural changes. From the theoretical formula of TDSF with the asymmetric structure, it was found that the asymmetric structure can effectively reduce the value of TDSF under the same design size parameters. With the same design parameters, the theoretical TDSF value of the symmetric structure was [−200.2 ppm/°C, −261.6 ppm/°C], and the theoretical TDSF value of the asymmetric structure was [−11.004 ppm/°C, −72.404 ppm/°C]. After that, the asymmetric structure was optimized by the vertical and horizontal halver method, and the optimal structure conforming to the laboratory processing conditions was obtained. Through analysis and checking, it was found that when β=5,s=15,d=10, adhesion will not occur. Finally, in the experiment, it was found that the structure designed in this work can be processed by HF dry etching and base wafer protection. The influence of the chip package on the signal output of the accelerometer was analyzed. The packaging measures effectively improved the linearity and stability of the output signal under high temperatures. The TDB before and after packaging was changed from 3000 to 60 μg/°C, and the TDSF was changed from 3000 to 140 ppm/°C. The design of the asymmetric structure was derived from the theory, and the theoretical advantage of the same parameter design was obtained in this work. In future work, it is necessary to further improve the optimized size and processing means of the asymmetric structure to obtain TDSF and TDB values closer to the ideal state.

## Figures and Tables

**Figure 1 sensors-19-03522-f001:**
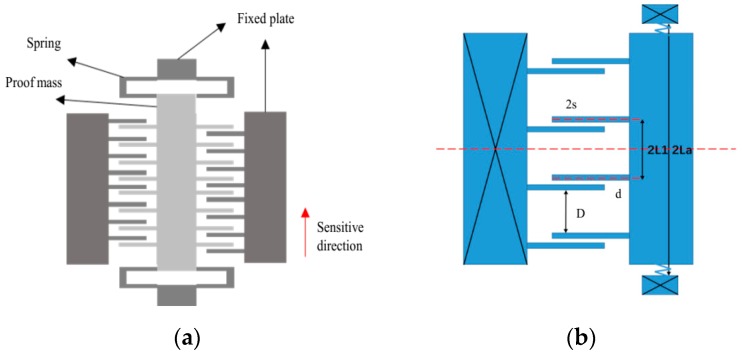
An asymmetric structure model: (**a**) overall structure diagram; (**b**) diagram of the asymmetric comb (d is the narrow gap, D is the wide gap, 2s is the width of the comb, $L_{a}$ is the distance from the anchors of the proof mass to the midline, $2L_{1}$ is the distance of two moving electrodes).

**Figure 2 sensors-19-03522-f002:**
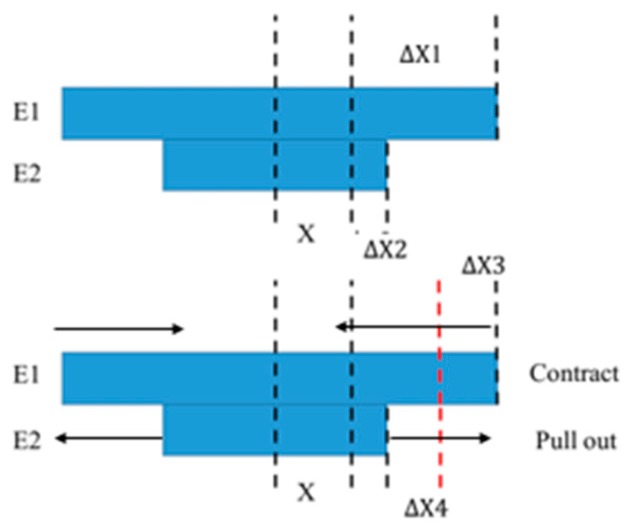
Diagram of the hypothesis of the expansion process of two materials.

**Figure 3 sensors-19-03522-f003:**
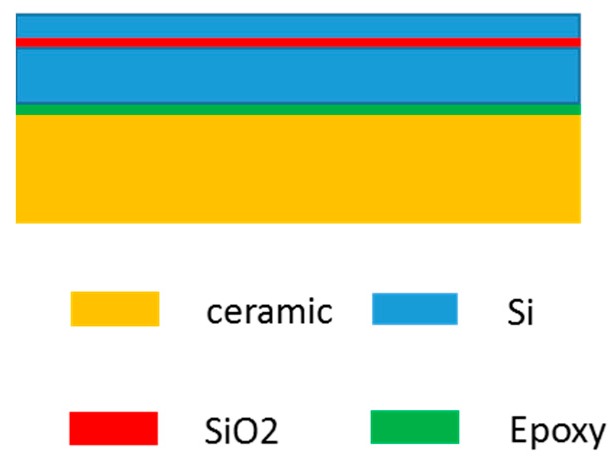
Five-layer model used for the numerical analysis.

**Figure 4 sensors-19-03522-f004:**
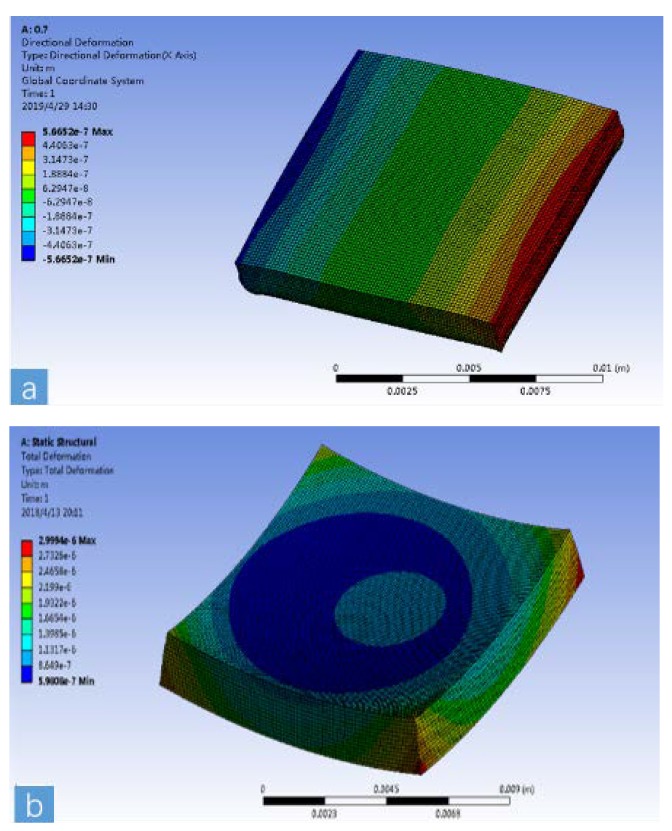
Two simulation results for a total silicon thickness of 335 μm: (**a**) fixed support at the bottom, constrained expansion; (**b**) free expansion, no boundary constraints.

**Figure 5 sensors-19-03522-f005:**
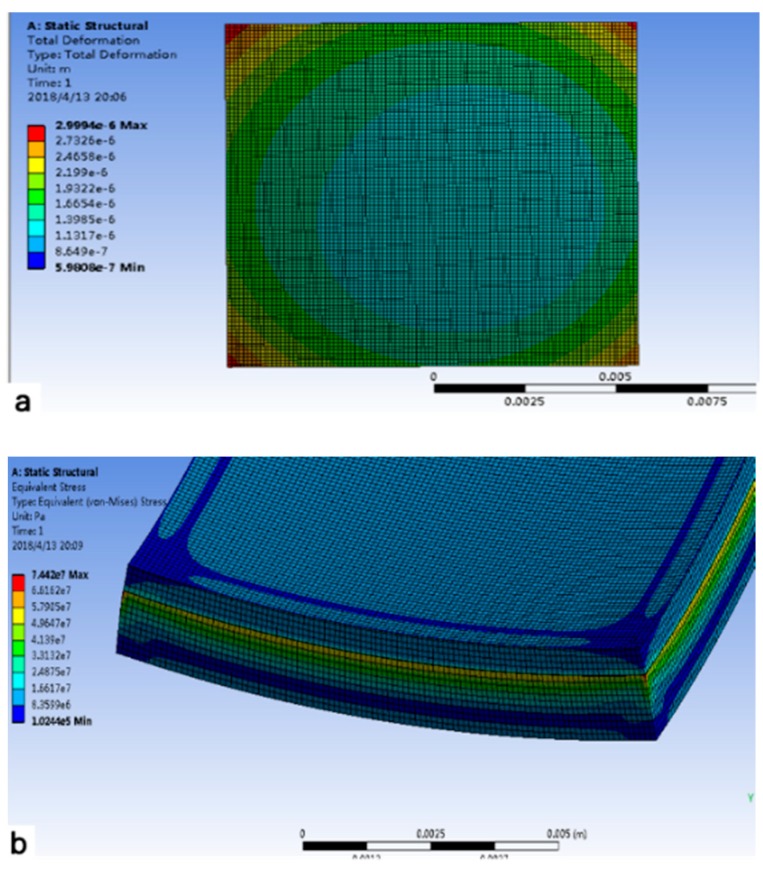
Stress distribution of the model during free expansion: (**a**) stress distribution at the bottom; (**b**) stress distribution on the sidewall.

**Figure 6 sensors-19-03522-f006:**
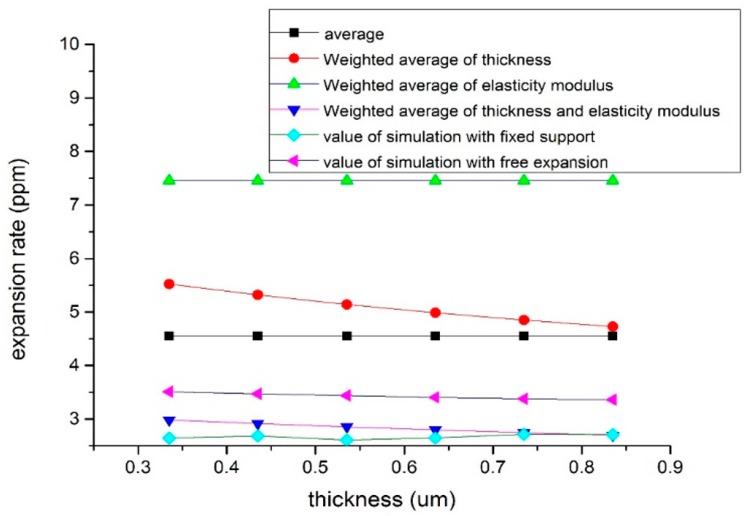
The change trend with the total thickness of silicon which is at the bottom of the SOI for the $\alpha_{eq}$ of the four theoretical calculation methods and two numerical simulation methods.

**Figure 7 sensors-19-03522-f007:**
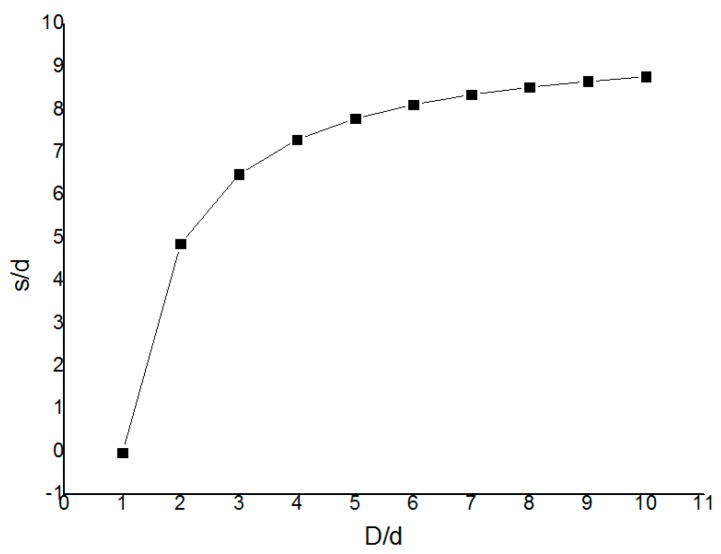
The relationship between s/d and β.

**Figure 8 sensors-19-03522-f008:**
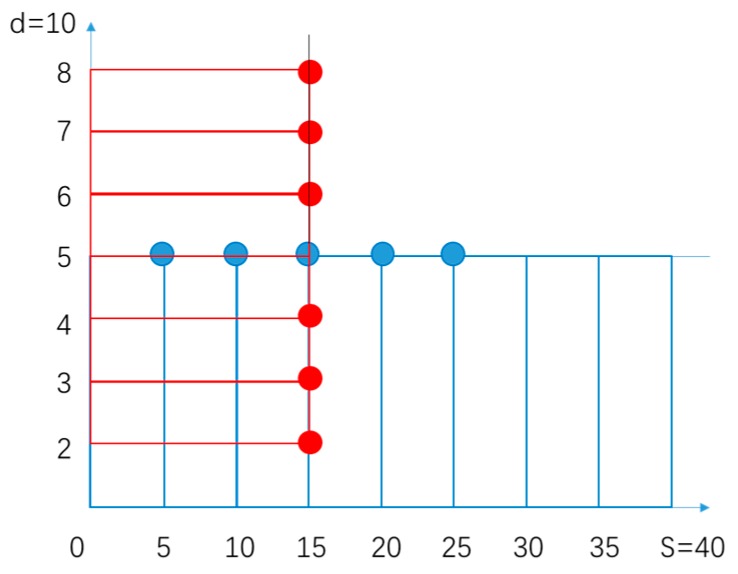
Principle of the “vertical and horizontal halver method”.

**Figure 9 sensors-19-03522-f009:**
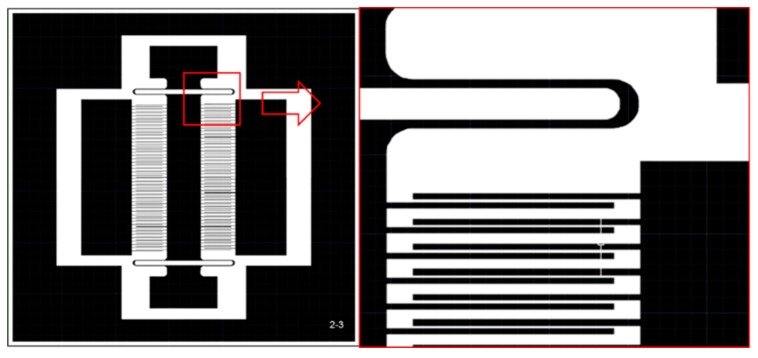
The optimized asymmetric structure design drawings.

**Figure 10 sensors-19-03522-f010:**
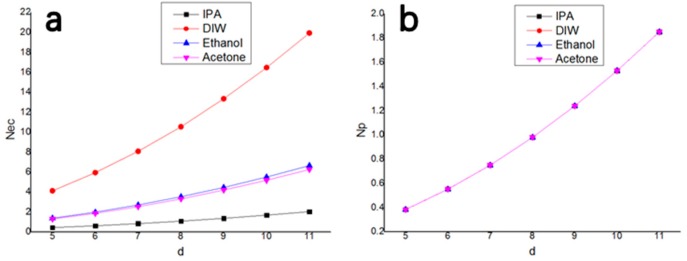
Trend diagrams of the dimensionless parameters $N_{EC}$ and $N_{P}$ with a small spacing *d*: (**a**) Trend diagram of $N_{EC}$ for four different solutions with *d*; (**b**) trend diagram of $N_{P}$ for four different solutions with *d*.

**Figure 11 sensors-19-03522-f011:**
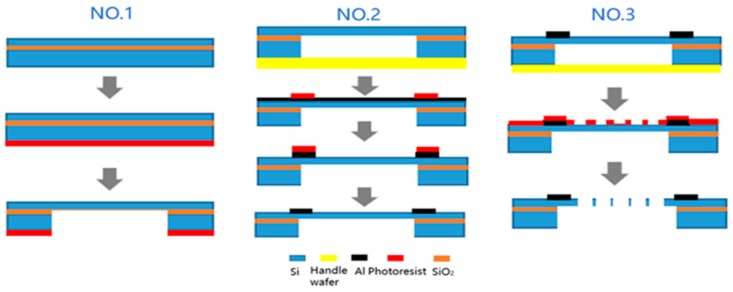
The processing scheme.

**Figure 12 sensors-19-03522-f012:**
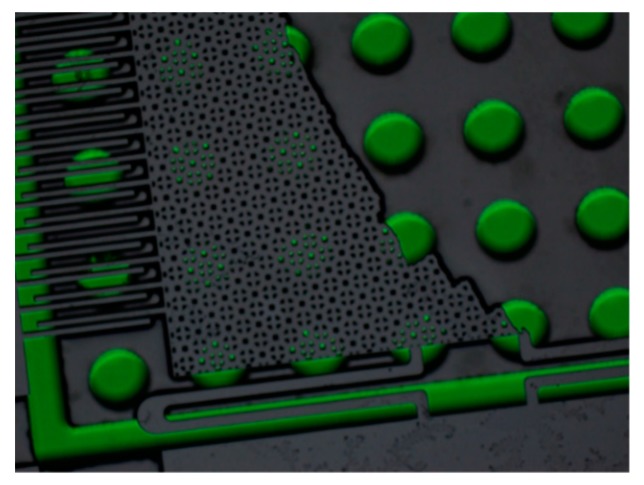
The proof mass and the bottom structure are stuck together.

**Figure 13 sensors-19-03522-f013:**
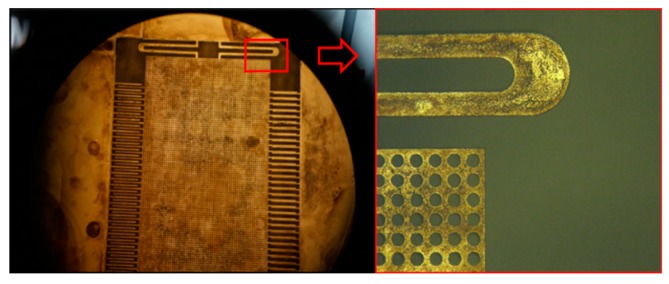
The final result of the combination processing of dry etching and wet etching.

**Figure 14 sensors-19-03522-f014:**
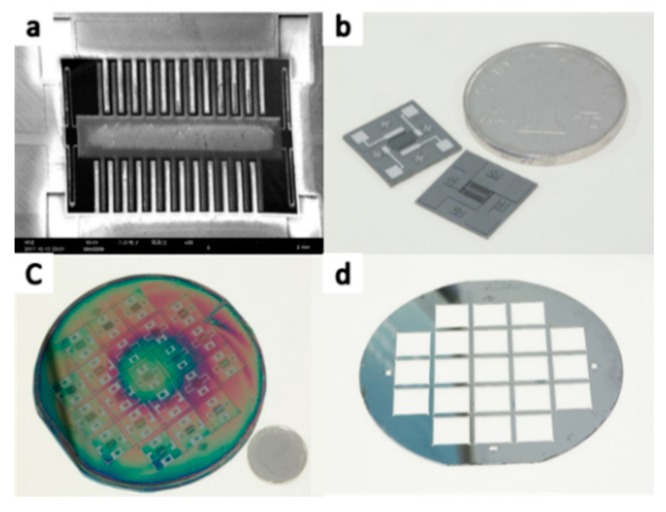
Picture of the asymmetric structure accelerometer: (**a**) the result of the SEM observation, (**b**) the small pattern taken by the camera, (**c**) the wafer that was not etched, (**d**) the frame that was etched away.

**Figure 15 sensors-19-03522-f015:**
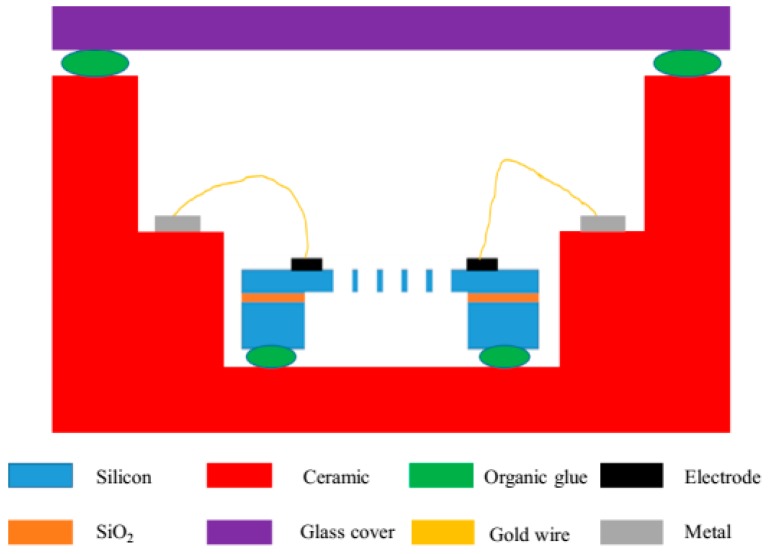
The structure of the package.

**Figure 16 sensors-19-03522-f016:**
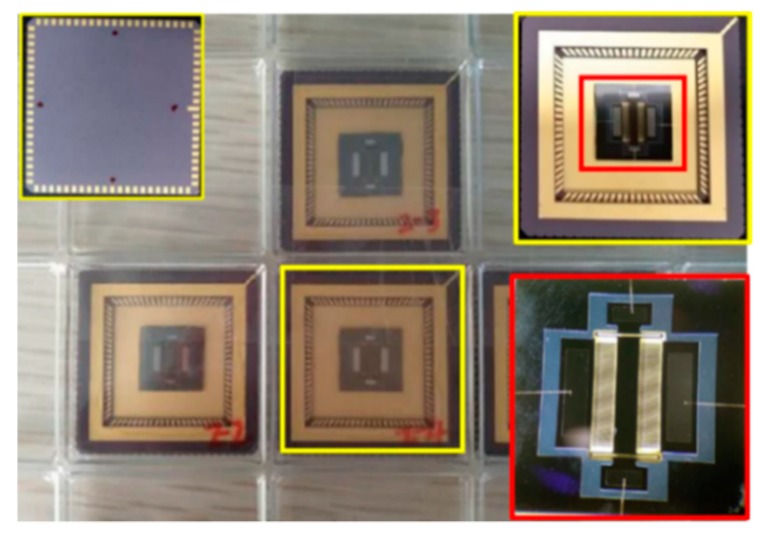
Photograph of the packaged chip.

**Figure 17 sensors-19-03522-f017:**
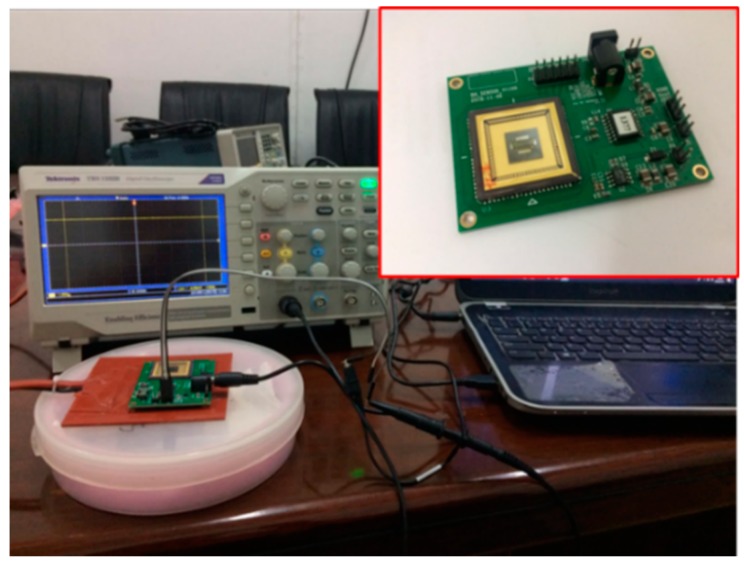
Packaged chip and circuit.

**Figure 18 sensors-19-03522-f018:**
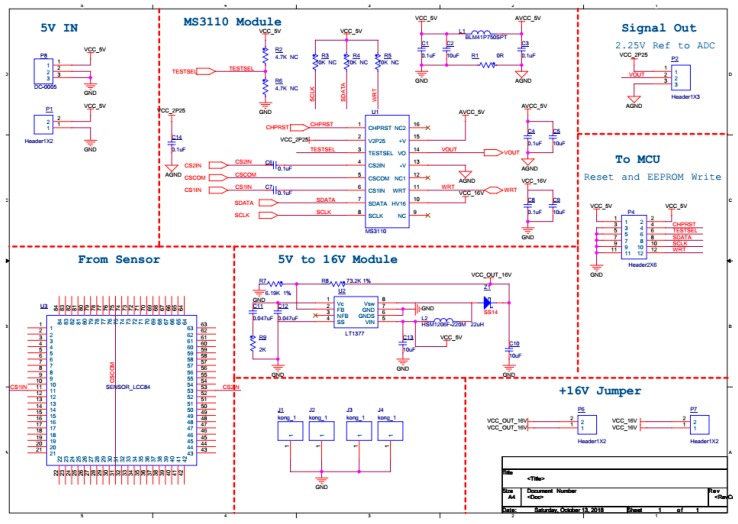
Signal readout and conditioning circuit.

**Figure 19 sensors-19-03522-f019:**
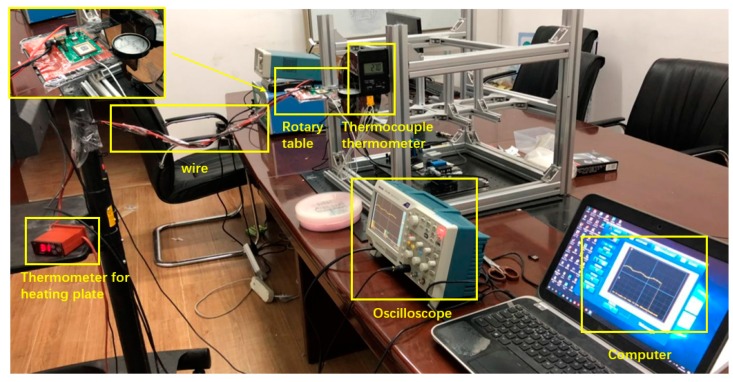
Experimental device.

**Figure 20 sensors-19-03522-f020:**
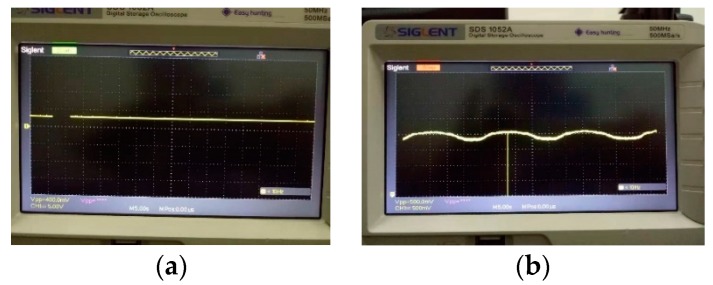
The output signal after the accelerometer was powered: (**a**) static signal output when the accelerometer was placed horizontally at room temperature; (**b**) sinusoidal signal output when the accelerometer rotated at low speed at room temperature (with a period of 16 s).

**Figure 21 sensors-19-03522-f021:**
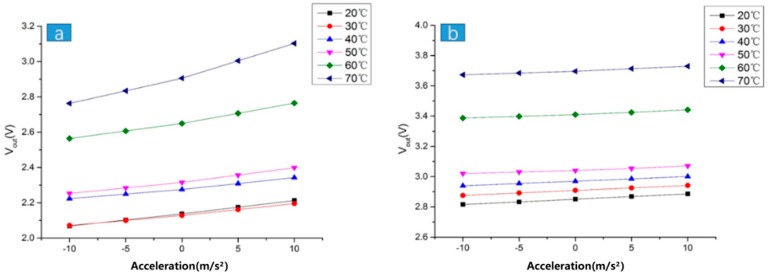
Relationship between the acceleration value and output voltage at different temperatures: (**a**) before packaging; (**b**) after packaging.

**Figure 22 sensors-19-03522-f022:**
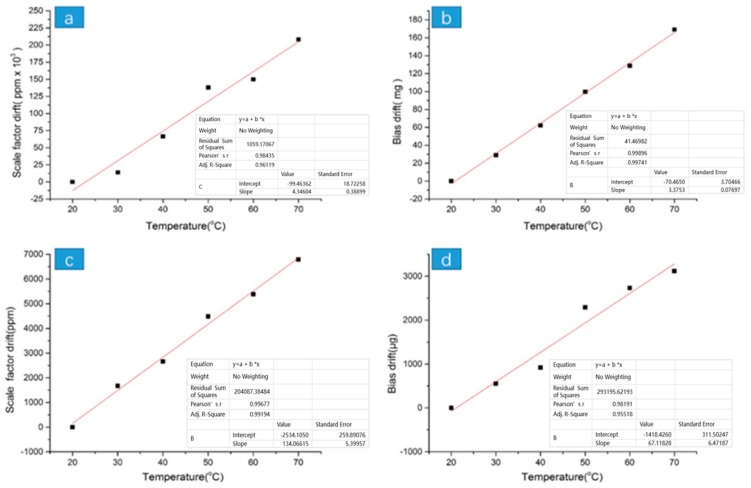
The test results of the scale factor drift and the bias drift before and after packaging: (**a**) the relationship between the scale factor drift and the temperature before packaging; (**b**) the relationship between the bias drift and the temperature before packaging; (**c**) the relationship between the scale factor drift and the temperature after packaging; (**d**) the relationship between the bias drift and the temperature after packaging.

**Figure 23 sensors-19-03522-f023:**
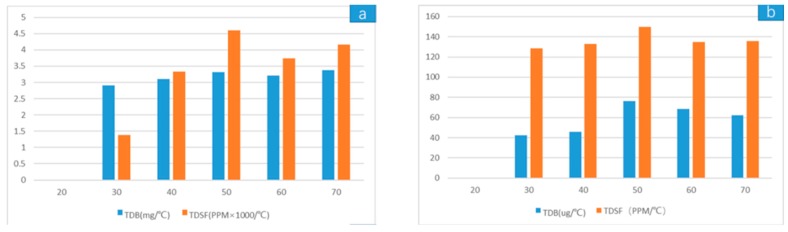
TDSF and TDB histograms about temperature distribution (**a**) Before packaging (**b**) After packaging.

**Table 1 sensors-19-03522-t001:** Physical properties of various materials.

Materials	Silicon	Ceramic	Epoxy	SiO_2_
Elasticity modulus (GPa)	190	400	2.5	170
Thickness (μm)	335~835	1000	4	1
CTE (ppm/°C)	2.6	6.5	60	0.8
Poisson’s ratio	0.28	0.22	0.38	0.16

**Table 2 sensors-19-03522-t002:** The optimized design structure size.

Property	Value	Units	Property	Value	Units
Folded beam width ($w_{k}$)	15	μm	Width of the proof mass ($w_{m}$)	2100	μm
Folded beam length ($l_{k}$)	1000	μm	Length of the proof mass ($l_{m}$)	6020	μm
Folded beam height ($h_{k}$)	35	μm	Height of the proof mass ($h_{m}$)	35	μm
Narrow capacitive gap (d)	10	μm	Width of the electrode ($w_{s}$)	30	μm
Wide capacitive gap (D)	50	μm	Length of the electrode ($l_{s}$)	900	μm
Length of the chip (L)	10000	μm	Height of the electrode ($h_{s}$)	35	μm
Width of the chip (W)	10000	μm	Number of fixed electrodes	28	-

**Table 3 sensors-19-03522-t003:** Flow and operation of the process.

Process Number	Process step	Operation
NO.1	Wafer Cleaning	Clean and dry the SOI wafer
	Photoresist coating	AZ4620, 3000 rpm, 50 s
	Pre-baking	95 °C, 60 min
	Exposure	TSK, 13 s
	Development	AZ400k: H_2_O = 1:3, 2 min
	Post-baking	95 °C, 30 min
	Etching	Etch protection ratio is 8:5–9:5, 300 μm
	Removal of photoresist	Piranha
	Removal of SiO_2_	HF
NO.2	Handle wafer	500 μm wafer at the bottom of SOI
	Sputtering	Direct target sputtering, Al, 300 s + 600 s
	Photoresist coating	S1813, 3000 rpm, 50 s, remove handle wafer
	Pre-baking	115 °C, 30 min
	Exposure	BSK, 3 s
	Development	NMD-3, 2 min
	Post-baking	115 °C, 15 min
	Corrosion	HF: Deionized water = 1:10, ≤1 min
	Removal of photoresist	Acetone, ultrasonic
NO.3	Photoresist coating	S1813, 3000 rpm, 50 s, remove handle wafer
	Pre-baking	115 °C, 30 min
	Exposure	BSK, 3 s
	Development	NMD-3, 2 min
	Exposure	BSK, 3 s
	Development	NMD-3, 2 min
	Etching	Etch protection ratio is 8:5, 25 μm
	Removal of photoresist	Acetone

**Table 4 sensors-19-03522-t004:** Results of the reference voltage test in four directions.

Vref-Up	Vref-Bottom	Vref-Left	Vref-Right
2.280928	2.280896	2.280768	2.280640
Range	0.000288	Relative error	0.012632%

**Table 5 sensors-19-03522-t005:** The test results of the temperature drift of bias (TDB) and temperature drift of scale factor (TDSF).

	TDB			TDSF		
	Absolute Value	Absolute Stability	Relative Stability	Absolute Value	Absolute Stability	Relative Stability
Before packaging	3 mg/°C	0.47 mg/°C	14.93%	3000 ppm/°C	3210 ppm/°C	93.20%
After packaging	60 μg/°C	33.69 μg/°C	57.016%	140 ppm/°C	20.97 ppm/°C	15.38%

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
