# Peer review of "Thermal Drift Investigation of an SOI-Based MEMS Capacitive Sensor with an Asymmetric Structure"

_sensors, 2019, doi:10.3390/s19163522_

Round 1

Reviewer 1 Report

In this paper, the authors describe the theoretical and experimental analysis of the temperature drift of an accelerometer. The concepts of an effective coefficient of thermal expansion and effective elasticity are proposed. In overall, the topic is interesting, however, the innovative content of the paper is marginally low.

Down below are only the most critical issues to be addressed by the authors.

The analyzed structure is not described properly. Suggest to alter the storyline of the paper and speak about the analyzed structure and technology prior to theoretical analysis.

Fig. 1 is copied from the previous publication and is not adequate for the present paper, because it is not explained properly which parts of the structure are being addressed by the analysis in the subchapter 2.1.

Fig. 2 does not provide any useful graphical information; it is impossible to interpret and understand it.

Numerical simulation in 2.1 is not described at all. Only the screenshots from the software and the line graph of the model outputs are shown. The simulation method itself (finite element analysis or something?) has to be described with sufficient details for the reader to check and repeat the analysis.

The heading of the Fig. 6 is not descriptive enough, because it does not explain what "thickness" in the horizontal axis means.

The generalization about the adequacy ("correctness") of the model as referred to an earlier publication is not adequate. Authors need to provide an explicit comparison of the model output with any reference or experiment and explain why they compare different structures (anodically bonded silicon/glass and adhesive bonded silicon/ceramics), which are incomparable.

It is not clear why subchapter 2.2 is needed. Does it introduce a different structure? How then it is related to the previous chapter?

The analytical reasoning in subchapter 2.3 is just a compilation of models that are published previously. Authors must stress their innovative approach instead of incrementing the concepts developed by someone else.

The design optimization in subchapter 2.4 describes common engineering practice and does not have any scientific value. Suggest to exclude this chapter or make it as brief as possible.

Testing part (subchapter 3.2) is missing the proper description and measurement diagram explaining how the chip is excited, how the measurement devices are connected, how the reference signals/values are controlled, how the measured data/information is processed, etc. Fig. 18 is not adequate and must to be replaced with the functional diagram of an experiment.

The oscillograms are to be disclosed and explained to the reader.

Fig. 20 does not provide any relevant information.

It is completely not clear what is the meaning of the voltage in the vertical axis of the Fig. 22.

Author Response

We are grateful to the reviewer for making such detailed comments and suggestions for our paper. We strongly agree with the comments and the suggestions, and carefully revise the paper in accordance with the comments and the suggestions. The attached file is point-to-point response. The highlighted red portions are the comments of the reviewer.

Reviewer 2 Report

It is an interesting calculation but there any many articles dealing with
simulation and measurements of MEMS capacitive sensors.
If the authors CLEARLY STATE what is their improvement and/or new
contribution then maybe could be publish. Please improve

Author Response

(The authors gave the same response as above.)

Round 2

Reviewer 2 Report

Thank you very much for considering my suggestions.  Good job

Author Response

Thank you very much for what you have done for us.